# A Review of Termite Species and Their Distribution in Thailand

**DOI:** 10.3390/insects13020186

**Published:** 2022-02-10

**Authors:** Watthanasak Lertlumnaphakul, Ratchadawan Ngoen-Klan, Charunee Vongkaluang, Theeraphap Chareonviriyaphap

**Affiliations:** 1Department of Entomology, Faculty of Agriculture, Kasetsart University, Bangkok 10900, Thailand; watthanasak.le@ku.th (W.L.); ngernklun@yahoo.com (R.N.-K.); 2Royal Forest Department, Bangkok 10900, Thailand; chaisavong@gmail.com

**Keywords:** termite species, survey, Blattodea, geographic information system, Thailand

## Abstract

**Simple Summary:**

There is little information available on termite species in Thailand. We reviewed termite species and information on their distribution in Thailand via the Google Scholar search platform and online databases with the QGIS program to create a map of termite species. In total, 44 coordinates were obtained from 19 reviewed sources, with approximately 75 species of termites identified at the species level and 83 unknown species of termites being reported across all regions.

**Abstract:**

Although 3105 termite species have been documented worldwide, little information is available on those in Thailand. In this review, the Google Scholar search platform and the Scopus and Science Direct databases were used to obtain information on termite species and for georeferencing. The QGIS software was used to create point localities that were overlaid on the Thailand administrative level 1 (province) to map the distribution of termite species in the country based on the World Geodetic System 1984. From the 19 reviewed sources, 44 defined coordinates were identified in 14 provinces across Thailand. Among these 44 coordinates, we found 75 termite species and 83 unknown species of termites; in total, 36 termite species were from the North (6 locations), 33 species were from the Northeast (10 locations), 34 species were from the West (4 locations), 29 species were from the Central region (3 locations), 44 species were from the East (8 locations), and 54 species were from the South (13 locations). The most predominant species in all regions were *Globitermes sulphureus*, *Macrotermes gilvus*, *Microcerotermes crassus*, and *Microtermes obesi*.

## 1. Introduction

Termites are an important group of eusocial insects and are among the most prevalent structural insect pests. As an important ecosystem insect, termites make up to 10% of animal biomass [1,2]. The activities of mound and gallery building by termite influence soil profile development, including the translocation of sub-surface soil to the surface and microbial structure. Termite activities have a significant impact on the porosity and water holding of soils and infiltration rates [3,4]. The decomposition of organic matter has been reported which results from the mutualistic interaction between fungi, bacteria, and termites to digest food [5,6,7,8]. This relationship is expected to have a positive effect on the decomposition of organic matter and nutrient cycling [4,9,10]. Termites’ success regarding organic matter decomposition and the effects of their activity on soil profile development, soil properties, and plant growth nutrient recycling rates have been well described [4]. Although termites are important beneficial insects, they cause major economic losses exceeding USD 40 billion annually [2,11,12]. They feed on ligno-cellulosic materials such as wood, wooden products, the structural wood of buildings, and furniture. In addition, they can destroy other materials that do not contain cellulose such as PVC pipes, plastics, and gypsum products [13].

Termites have been surveyed in many tropical and subtropical regions worldwide, and many termite species have been identified. Thailand is a tropical country with various types of forest appropriate for the growth and development of termites [14]; however, there is limited information on the termite species and their ecology in Thailand, except for the earliest surveys conducted by Holmgren [15], who reported five termite species, namely, *Bifiditermes indicus* (Holmgren), *Glyptotermes domesticus* (Havilandi), *Coptotermes havilandi* (Holmgren), *Macrotermes carbonarius* (Hagen), and *Odontotermes formosanus* (Shiraki). Later, *Macrotermes annandalei* was identified in Thailand and added to the catalogue of termites of the world by Snyder [16]. In 2004, the termites of Thailand were classified into 4 families: Kalotermitidae (1 subfamily, 6 genera), Archotermopsidae (1 subfamily, 1 genus), Rhinotermitidae (4 subfamilies, 5 genera), and Termitidae (4 subfamilies, 27 genera) [17,18]. Ahmad [19] published a monograph on the termite species of Thailand that became a standard identification key. Subsequently, Morimoto [20] published a termite identification key named “Termite from Thailand”, and 48 species were added to the key. Sornnuwat et al. [18] published the most recent identification key of termites in Thailand with 178 species in 37 genera from 10 subfamilies and 4 families from 53 Thai provinces. Between 1965 and 2021, over 200 termite species in Thailand were recorded belonging to 39 genera in 10 subfamilies and 4 families. These studies have contributed immensely to the knowledge on termites in the country; however, compared to other countries, there has been little investment in research facilities such as termite laboratories and field stations, and in collecting information regarding the distributions of species throughout the country. Thus, the review herein was conducted to provide more detailed information on species distribution in different regions of Thailand. The obtained results contribute to the knowledge base that can be disseminated broadly and may also be used in others related research.

In this review, we compile information on the termite species in Thailand formerly published between 1965 and 2021 using online platforms and databases. The presentation was based on a geographic information system (GIS) to create distribution maps of the known termite species in the country using the QGIS software.

## 2. Materials and Methods

The information on species and the distribution of termites in Thailand between 1965 and 2021 was searched using Google Scholar; the website of the Forest Research and Development Office, Royal Forest Department; Scopus; Science Direct. The data were combined to create a GIS map of termite species reported in Thailand. Key words included “termite and Thailand”, “termite and survey and Thailand”, “termite or Isoptera and Thailand”, “rhinotermitidae and Thailand”, “termites and Thailand”, “termite and species and Thailand”, and “termitidae and Thailand”. Only termites identified at the genus or species level with GIS coordinates were included.

### 2.1. Selection Criteria

#### 2.1.1. Publications in Database

Three selection criteria were used to select relevant publications: (1) studies on termite species carried out in Thailand with known geographical coordinates; (2) year of publication between 1965 and 2021; and (3) termite identification at the genus or species taxon level. All publications were obtained using Google Scholar or the Scopus or Science Direct databases (Figure 1).

#### 2.1.2. Annual Reports/Theses

Annual reports and theses on termite species were obtained from the online website of the Royal Forest Department, Bangkok, Thailand, the online library of Kasetsart University, Bangkok, Thailand, or the Thai Digital Collection database (TDC) under the Thai Library Integrated System (Thai LIS). However, all termite information was selected based on the same criteria described above (Figure 1).

### 2.2. GIS Map Construction

The Thailand Subnational Administrative Boundaries level 1 (province) dataset was derived from https://data.humdata.org/dataset/thailand-administrative-boundaries, accessed on 18 May 2021, based on the World Geodetic System 1984 (WGS 84)-EPSG 4326. Georeferenced locations of recorded termite species throughout the country were tabulated in a CSV file (Comma delimited: *.csv) and converted to point data in QGIS (desktop version 3.10.11) (open source software available under the terms of the GNU General Public License, Copyright (C) 1989, 1991, Free Software Foundation, Inc., 59 Temple Place-Suite 330, Boston, MA 02111-1307, USA; https://www.qgis.org, accessed on 18 May 2021). Each point datum represented an identified termite species, and the names of districts, provinces, and regions were overlaid on the Thailand Subnational Administrative Boundaries level 1 (province). To display many species of termites at the same location, the Point Displacement Renderer was applied to visualize all features of a point layer. Briefly, the symbols of the points were arranged on a displacement circle around a center symbol at the georeferenced location. A six-region system, formalized in 1977 by the National Geographical Committee, was used: North, Northeast, West, Central, East, and South.

## 3. Results

Only 19 studies matched the selection criteria. Of these 19 studies, eight were peer-reviewed publications, eight were national or country reports, and the remaining three were graduate student theses (Figure 1), making up a total of 44 definite geographic coordinates across the country. The surveyed sites were in 14 provinces in 6 regions across Thailand (Figure 2, Figure 3, Figure 4, Figure 5, Figure 6 and Figure 7). There were 75 identified termite species and approximately 83 unknown species of termites, for a total of 158 termite species. In the Northern region, 36 termite species belonging to 19 genera were identified from Chiang Mai (five locations) and Lampang (one location), as shown in Figure 2. In Northeastern Thailand, 33 species in 20 genera were recorded from Nakhon Ratchasima (nine locations) and Khon Kaen (one location), as shown in Figure 3. In Western Thailand, 34 species classified into 17 genera were identified in Tak (one location), Kanchanaburi (two locations), and Prachuap Khiri Khan (one location), as shown in Figure 4. A total of 29 species of termites belonging to 20 genera were identified from Phitsanulok (two locations) and Saraburi (one location) in Central Thailand (Figure 5). In the Eastern region, 44 species of termites belonging to 20 genera were identified from eight locations in Chanthaburi Province (Figure 6). The Southern region of Thailand was surveyed and 54 termite species in 23 genera were identified, of which 35 species were from the mainland (three locations) in Trang and Songkhla, and 37 species were from islands (10 locations) in Phang Nga and Chumphon provinces (Figure 7). Among the 28 commonly predominant termite species found, only *G. sulphureus*, *M. gilvus*, *M, crassus*, and *M. obesi* were reported in all regions in Thailand. Furthermore, 13 termite species were found in five regions and 7 species were recorded from four regions (Table 1).

In 1965, 64 termite species were described and classified into 29 genera from approximately 400 colonies, with 32 out of 79 species being new reports [19]. In 1997, 10 species and 13 species were found at fire-protected and unprotected sites, respectively, at Doi Suthep-Pui National Park in Chiang Mai Province, Northern Thailand. The fire-prone site had a greater number of Macrotermitinae species than the fire-protected site [21]. Amornsak et al. [22] found 20 termite species and two unidentified species in the hill evergreen forest of Khao Kitchakut National Park in Chanthaburi Province. Two undescribed species in the genera *Hospitalitermes* and *Bulbitermes* were new records. Later, the first record in Thailand of *Microcerotermes serrula* (Desneux) was collected from an aerial nest in Songkhla Province [23]. Another study of the species and the abundance of termites in different forests in Khao Kitchagoot National Park, Chantaburi Province, found 30 termite species belonging to 18 genera [24]. From this study, *Ancistrotermes pakistanicus* (Ahmad) was the dominant species at low altitudes, while *B. parapusillus* (Ahmad) and *Nasutitermes matangensis* (Haviland) were dominant at high altitudes [24]. Inoue et al. [25] studied termite species in the dry evergreen forest at the Sakaerat Environmental Research Station in November 1998 and February 1999 and reported 23 termite species in 17 genera, with no significant difference in termite abundance and biomass between the two sampling periods. In 2009, Choosai et al. [26] studied the termite species in the mounds and dykes of paddy fields in Baan Daeng village, Ban Fang District, Khon Kaen Province, in Northeastern Thailand. The macrofauna at five different types of location were sampled, and the soil macro-invertebrates were collected; termite specimens were classified and identified to the species level. The results indicated that there were four species of soil- and litter-feeding termites and one species of soil-feeding termite in mounds and dykes, respectively. The related termites found were *Odontotermes formosanus* (Shiraki), *H. ataramensis* (Prashad & Sen-Sarma), *Angulitermes* sp., *Microcerotermes* sp., and *Pericapritermes* sp., but *O. formosanus* was the dominant species that was recorded in all mounds [26]. Shaleh et al. [27] studied fungus-growing termite colonies and mutualistic fungi (*Termitomyces* sp.) collected from four termite colonies at the Sakaerat Biosphere Research Station in Nakhon Ratchasima Province and from two colonies at the Chulabhorn Dam in Chaiyaphum Province in Northeastern Thailand. The termites were morphologically identified and compared using molecular identification to confirm both identification procedures at the genus level were consistent. *Macrotermes annandalei* (Silvestri), *A. pakistanicus* (Ahmad), and *Odontotermes feae* (Wasmann) were identified from the Sakaerat colonies using both methods. *Ancistrotermes* sp. from the Chulabhorn Dam colonies was also identified based on both features [27] (Appendix A).

Kuntha et al. [28] reported a termite survey on the Khon Kaen University campus, with nine termite species belonging to six genera in two families being identified. Only *Coptotermes gestroi* (Wasmann) was listed in the family Rhinotermitidae, whereas the other eight species comprised *M. gilvus* (Hagen), *Microtermes pakistanicus* (now *A. pakistanicus* (Ahmad)), *M. obesi* (Holmgren), *Microcerotermes crassus* (Snyder), *O. feae*, *Odontotermes longignathus* (Holmgren), *O. formosanus*, and *G. sulphureus* (Haviland) and were classified into the family Termitidae [28]. Vongkaluang et al. [29] studied the species of termites in four different forests in Chanthaburi and Kanchanaburi Provinces. They identified 34 termite species in 18 genera from moist evergreen forest, 27 species in 15 genera from dry evergreen forest, 23 species in 13 genera from hill evergreen forest, and 35 species in 13 genera from secondary dry dipterocarp forest. In addition, Termopsidae, an extinct termite family in Blattodea, was reported, and 11 genera were described as new genera. Additionally, termite species on the islands of Ko Phra-Thong and Ko Ra in Phang Nga Province were surveyed in April 2004 [30], and 25 termite species belonging to 15 genera in 6 subfamilies were identified from Ko Phra-Thong, while 29 species were found in Ko Ra and classified into 16 genera and 6 subfamilies. The wood-feeding termite *M. crassus* was the predominant species that was found on both islands [30]. Sornnuwat [31] studied termites from five national parks in 2000 and found that Jae-Sorn National Park had the greatest number of termite species (35), followed by Doi Suthep-Pui, Mae-Takhrai, Aub-Luang, and Doi Inthanon. Furthermore, *Odontotermes* was the main genus found at almost all study locations, but the family Rhinotermitidae was not found in Doi Inthanon National Park. Sornnuwat et al. [32] recorded 35 species in 13 genera from secondary dry dipterocarp forest (SDDF) and 17 species in 11 genera from dry evergreen forest (DEF) in 2001. Macrotermitinae was the major group playing an important role in SDDF, while Termitinae and Macrotermitinae had a major role in DEF. *Odontotermes* sp. was reported as the dominant species in both forests. In addition, termites on the islands of Koh Khai and Koh Wiang in Pathio District in Chumphon Province were characterized in 2001, and 6 and 18 species, respectively, were identified. *Odontotermes proformosanus* (Ahmad) and *M. crassus* were the dominant species on both islands [33] (Appendix A).

Furthermore, Teekatananont [34] studied termite species in fruit orchards in the Khao Kitchakut Subdistrict in Chanthaburi Province, Eastern Thailand, where the major fruits being grown were durian, rambutan, and mangosteen. The transect method was applied to survey the termite species. Of the nine termite species identified, only six species were in the rambutan orchard, namely, *M. pakistanicus* (now *A. pakistanicus* (Ahmad))*, O. formosanus, Nasutitermes matangensiformis* (now *N. matangensis* (Haviland))*, Dicuspiditermes makhamensis* (Ahmad)*, G. sulphureus,* and *Termes cosmis* (Haviland). Four species (*Macrotermes carbonarius* (Hagen), *M. pakistanicus*, *O. formosanus*, and *G. sulphureus*) were in the durian orchard, and three species (*M. pakistanicus*, *M. obesi*, and *M. crassus*) were in the mangosteen orchard [34]. In 2010, the Plant Genetic Conservation Project under the Royal Initiative of Her Royal Highness Princess Maha Chakri Sirindhorn (RSPG) conducted inspections on six islands of the Mu Koh Similan National Park and reported 21 termite species in 15 genera. Ko Similan had the most termites with 18 species in 15 genera followed by Ko Payang (13 species in 10 genera), Ko Huyong (12 species in 8 genera), Ko Payu (12 species in 10 genera), Ko Miang (11 species in 9 genera), and Ko Bangu (11 species in 9 genera) [35]. Anan et al. [36] reported the species of termites and the seasonal effect on species on the campus of the Rajamangala University of Technology Thanyaburi (RMUTT), Patumthani Province. All specimens were classified into six genera, namely, *Odontotermes*, *Macrotermes*, *Microcerotermes*, *Termes*, *Coptotermes*, and *Parrhinotermes* (Appendix A).

Thipsantia [14] studied the biospecies of termites in two forest types (dry dipterocarp forest and dry evergreen forest) at the Sakaerat Environmental Research Station, Nakhon Ratchasima Province, from October 2009 to September 2010. There was a greater number of termite species in the dry evergreen forest compared to the dry dipterocarp forest, with 25 species belonging to 18 genera and 18 species belonging to 14 genera, respectively. *M. crassus* was the most dominant species in both forest types, while the subfamilies Kalotermitinae and Rhinotermitinae were only found in dry evergreen forest. Another termite survey was performed on two forest types (dry dipterocarp forest and mixed deciduous forest) at the Nongrawiang Center in Nakhon Ratchasima Province by Sookruksawong et al. [37]. Direct search, soil pit, and bait sampling methods were used to collect specimens that were identified to the species level. They identified 2 Termitidae subfamilies in the dry dipterocarp forest, consisting of 9 species in 5 genera, while 3 subfamilies and 2 families, comprising 11 species in 7 genera, were identified in the mixed deciduous forest. *Dicuspiditermes* sp. and *C. gestroi* were only identified in the mixed deciduous forest [37] (Appendix A).

## 4. Conclusions

From 1965 to 2021, although many articles studied termite fields in Thailand, only 19 articles presented information on termite surveys that included geographical coordinates. The coordinate data from these 19 publications were used to create a new GIS map showing the distribution of termite species across regions. There were 44 surveyed sites in 14 provinces in 6 regions, and 75 identified termite species and approximately 83 unknown species of termites, totally 158 termite species. were distributed throughout the country. However, ongoing termite surveys are required to update the termite species and their distribution status. In addition, not only surveys but also identification keys should be updated, and we recommend that future research should also focus on termite biology and ecology.

## Figures and Tables

**Figure 1 insects-13-00186-f001:**
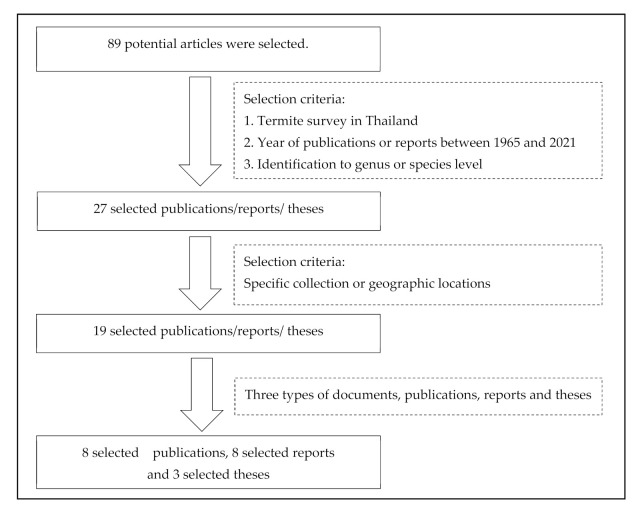
Chart of selection process following selection criteria.

**Figure 2 insects-13-00186-f002:**
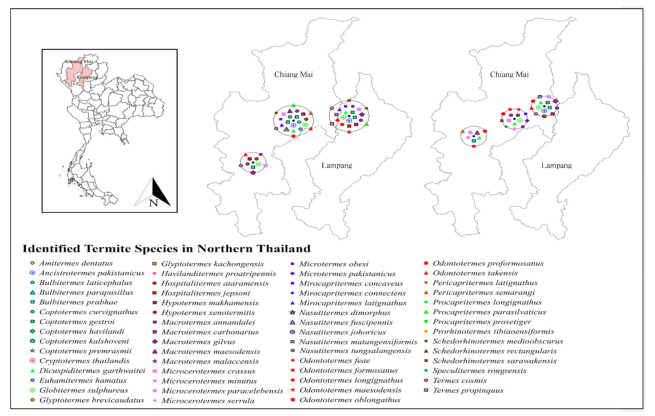
Distribution map of identified termite species in Northern Thailand based on previous studies between 1965 and 2021.

**Figure 3 insects-13-00186-f003:**
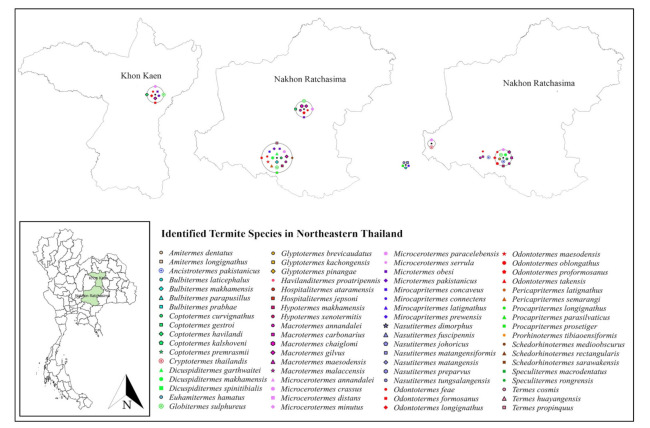
Distribution map of identified termite species in Northeastern Thailand based on previous studies between 1965 and 2021.

**Figure 4 insects-13-00186-f004:**
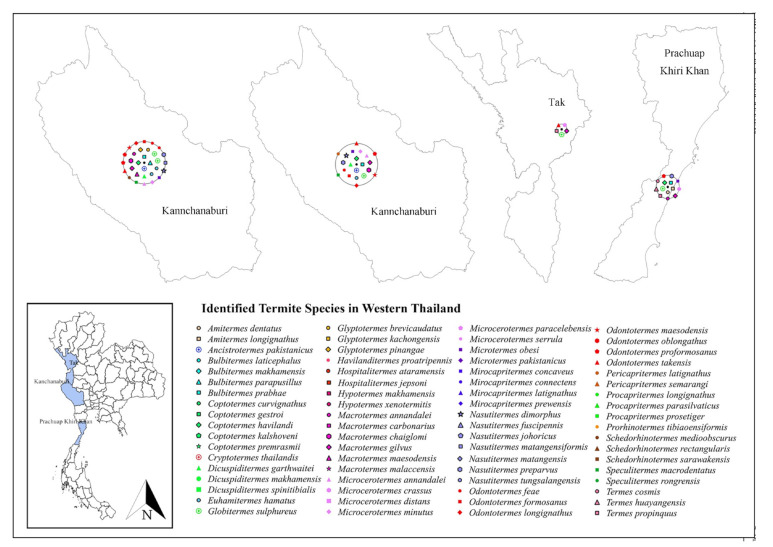
Distribution map of identified termite species in Western Thailand based on previous studies between 1965 and 2021.

**Figure 5 insects-13-00186-f005:**
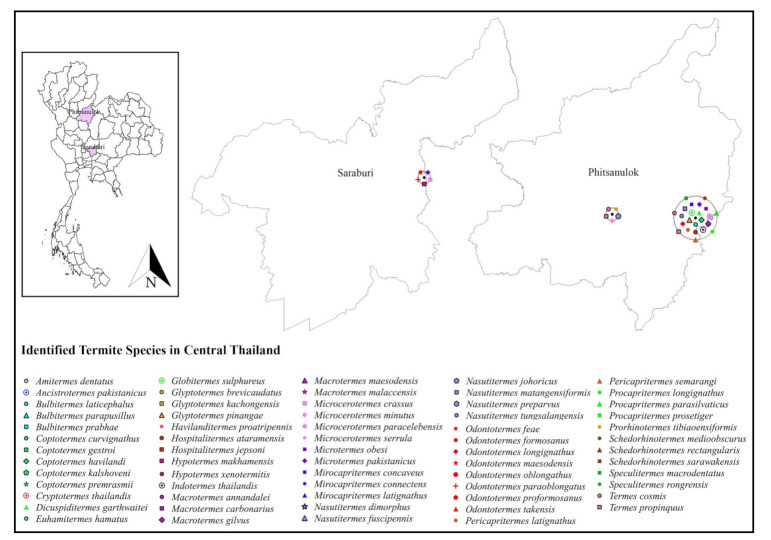
Distribution map of identified termite species in Central Thailand based on previous studies between 1965 and 2021.

**Figure 6 insects-13-00186-f006:**
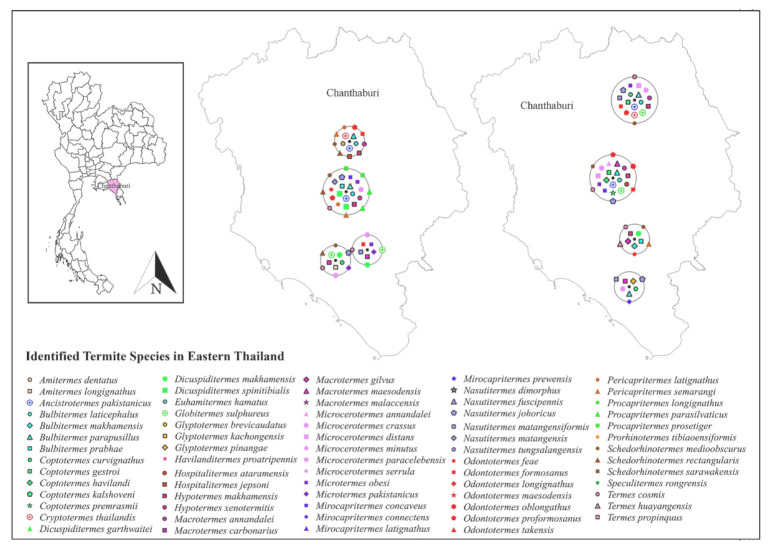
Distribution map of identified termite species in Eastern Thailand based on previous studies between 1965 and 2021.

**Figure 7 insects-13-00186-f007:**
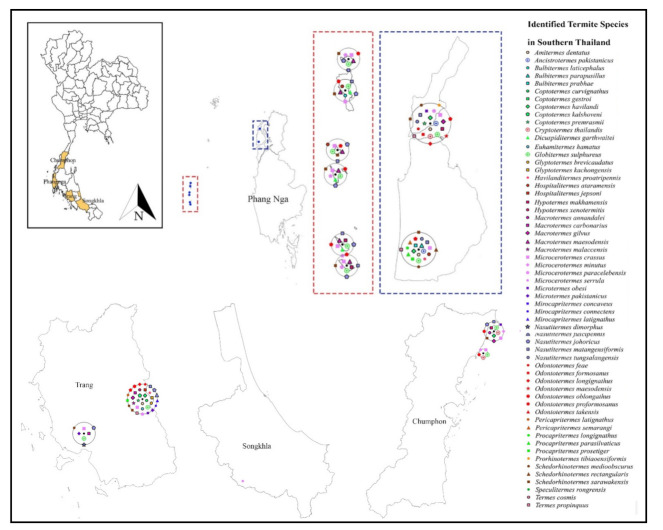
Distribution map of identified termite species in Southern Thailand based on previous studies between 1965 and 2021.

**Table 1 insects-13-00186-t001:** List of 28 most common termite species of Thailand and their distributions.

Species	Regions					
	Northern	Northeastern	Central	Eastern	Western	Southern
*Ancistrotermes pakestanicus*	✓	✓		✓	✓	✓
*Bulbitermes laticephalus*	✓	✓	✓	✓		✓
*Bulbitermes prabhae*	✓			✓	✓	✓
*Coptotermes curvignathus*	✓	✓		✓		✓
*Coptotermes gestroi*	✓	✓		✓	✓	✓
*Cryptotermes thailandis*				✓		✓
*Dicuspiditermes garthwaitei*	✓	✓			✓	✓
*Globitermes sulphureus*	✓	✓	✓	✓	✓	✓
*Glyptotermes pinangae*			✓	✓	✓	
*Hospitalitermes ataramensis*	✓	✓	✓	✓		✓
*Hypotermes makhamensis*	✓	✓	✓	✓		✓
*Macrotermes annandalei*	✓	✓		✓	✓	✓
*Macrotermes carbonarius*		✓		✓		✓
*Macrotermes gilvus*	✓	✓	✓	✓	✓	✓
*Macrotermes maesodensis*	✓			✓	✓	✓
*Microcerotermes crassus*	✓	✓	✓	✓	✓	✓
*Microcerotermes minutus*	✓		✓		✓	✓
*Microtermes obesi*	✓	✓	✓	✓	✓	✓
*Nasutitermes matangensiformis*	✓		✓	✓	✓	✓
*Odontotermes feae*	✓	✓		✓	✓	✓
*Odontotermes formosanus*	✓		✓	✓	✓	✓
*Odontotermes longignathus*	✓	✓	✓		✓	✓
*Odontotermes proformosanus*	✓	✓		✓	✓	✓
*Pericapritermes latignathus*				✓	✓	✓
*Pericapritermes semarangi*	✓	✓	✓	✓		✓
*Procapritermes parasilvaticus*	✓		✓	✓		✓
*Schedorhinotermes medioobscurus*	✓	✓		✓	✓	✓
*Termes cosmis*	✓		✓	✓	✓	

## Data Availability

Detailed data are available upon request.

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
