# Peer review of "A Review of Termite Species and Their Distribution in Thailand"

_insects, 2022, doi:10.3390/insects13020186_

Round 1

Reviewer 1 Report

A good revision.  The resolution of the figures was quit lower than in the first manuscript. Keep the resolution.

Author Response

A good revision. The resolution of the figures was quite lower than in the first manuscript. Keep the resolution.

We appreciated for your comment. However, the figures were increased the resolution following the attached file.

Reviewer 2 Report

I am reviewing this manuscript for the second time.  Oddly, the pdf of the manuscript indicates comments by at least one other reviewer.

-I found this version of the manuscript easier to read.

-Line 103, “potin” should be “point” correct?

-“The dates examined were 1913-2021.  This range is too wide for the possible papers, as GIS was not really developed until 1963 by Tomlinson.  This may be a problem due to unclear methods.  Were estimates used when known geographic locations were found in the papers (say a collection from a given town in older papers could be assigned current GIS location data for that town in 2021)?  This would allow for older papers to be included more easily.  “

This consideration was pointed out in my previous review.  Were pre-1963 papers included, or did they get removed due to the lack of coordinates?  If so, then please correct the years of articles searched to 1963-2021.

-“While part of this method was to use the internet search engines to find papers of interest and filter them as in figure 1, this method runs a good chance of passing over older papers which have not been catalogued yet (or improperly).  Part of doing a good review is finding these hidden papers.  Were the search engines used exclusively to find papers, or were other methods included?  I guess the question here is how good a job could the search engines do by themselves, or was this a more traditional review paper (using references within papers to extend the search to previous work and beyond)?”

Again from my prior review.  It seems that the authors used only the internet search engines to find potential papers.  As such it seems this manuscript is more of a test of internet search engines’ Thailand-termite capacity than a real review of papers in the area.

-As an older reader, I would appreciate the authors spending some time making the geographical figures more easily read.

-I definitely agree with one of the prior reviewers that if the authors know the number of termites from the combined regions, please just use that number (add up the ones in the abstract).

Author Response

- I am reviewing this manuscript for the second time. Oddly, the pdf of the manuscript indicates comments by at least one other reviewer.

- I found this version of the manuscript easier to read.

-As an older reader, I would appreciate the authors spending some time making the geographical figures more easily read.

Because of the English language proof-reading, we still attached the revision from native English expert who revised my manuscript before resubmitted to INSECTS journal. Now, we remove those comments also. However, we appreciated for your comment.

-Line 103, “potin” should be “point” correct?

Thank you for your comment. We have revised it.

- “The dates examined were 1913-2021.  This range is too wide for the possible papers, as GIS was not really developed until 1963 by Tomlinson.  This may be a problem due to unclear methods.  Were estimates used when known geographic locations were found in the papers (say a collection from a given town in older papers could be assigned current GIS location data for that town in 2021)?  This would allow for older papers to be included more easily.”

- This consideration was pointed out in my previous review.  Were pre-1963 papers included, or did they get removed due to the lack of coordinates?  If so, then please correct the years of articles searched to 1963-2021.

Thank you, sir. Actually, the first report of termite species in Thailand was published in 1913 without the geographical coordinates. Following your suggestion, we narrowly scoped about the year which we collected data. So, the year which occurred in this manuscript is in 1965-2021.

- “While part of this method was to use the internet search engines to find papers of interest and filter them as in figure 1, this method runs a good chance of passing over older papers which have not been catalogued yet (or improperly).  Part of doing a good review is finding these hidden papers.  Were the search engines used exclusively to find papers, or were other methods included?  I guess the question here is how good a job could the search engines do by themselves, or was this a more traditional review paper (using references within papers to extend the search to previous work and beyond)?”

- Again, from my prior review.  It seems that the authors used only the internet search engines to find potential papers.  As such it seems this manuscript is more of a test of internet search engines’ Thailand-termite capacity than a real review of papers in the area.

We used both traditional review paper (using references within papers to extend the search to previous work and beyond) and the internet search engines to find papers that were hidden. For example, some articles are local report that published in Thai organization or Thai website so we used the internet search engines to explore them and collect those data.

- I definitely agree with one of the prior reviewers that if the authors know the number of termites from the combined regions, please just use that number (add up the ones in the abstract).

We followed your suggestion. From all coordinates, there were 75 identified species of termites and approximately 83 unknown species of termites, totally 158 termite species found.

Round 2

Reviewer 2 Report

I appreciate the work the authors have put into this manuscript both in the original form and in answering reviewers' questions.

This manuscript is a resubmission of an earlier submission. The following is a list of the peer review reports and author responses from that submission.

Round 1

Reviewer 1 Report

P1 Simple Summary

It can be written more simply. The specific figures for each region are also given in the abstract and can be omitted here. It could be simpler, e.g. " more than ### species reporated from all regions...".

P2 L44-L58

There is no explanation as to why there is so little information of termite species in Thailand. There is also no comparative evidence for the paucity. Is there less termite research in Thailand than in other countries/regions within tropics? Or is it that there is less termite research not only in Thailand but also, for example, in South-East Asia than elsewhere? And in either case, what do the authors think are the reasons for the paucity of research? Such background information would ensure the importance of your research, but is not provided by the authors. Please add a brief explanation in the Introduction, adding citations where necessary, of any information that makes it possible to say that Thailand is particularly poorly researched and why there is so little.

Figures 2—7

The mapping legend is very confusing. Symbols for the same species should be same (same shape and color) throughout all figures (Fig2 to Fig7). The shape and color of the symbols should not be randomly designed, but should be specified, for example, the same color should be used for the same termite family. The elements of the figure should be designed to facilitate the reader's understanding.

P11 Conclusions

The first half of the Conclusions (the first four sentences) is inappropriate as it is not conclusion but a mere fact. In particular, the third sentence, which describes about higher and lower termites, is completely out of context with the preceding and following sentences; the first four sentences are not conclusions but background information, and fairly general background information. The second half of the paper is also very brief, stating results rather than conclusions, and the authors' arguments are unclear. For example, it is not clear how the authors conclude that there were only eight papers that could be used for geo-referencing analysis: are you saying that eight studies are not enough, or are you saying that eight studies provide enough information for geo-referencing analysis? The authors should be encouraged to develop an argument for how the results of this study can be positioned.

P2 L49 […by [15].]

P9 L384 […by [37].]

The reference number should not be a sentence constituent. Therefore, these are incomplete sentences.

P9L388 There is a period missing from sp.

Reviewer 2 Report

This was an interesting idea showing the value of cross-referencing animal collections with proper geographical data to estimate termite species found in regions of Thailand.

However, some problems are noted. 

-The English language of the manuscript will need moderate editorial help.

-The dates examined were 1913-2021.  This range is too wide for the possible papers, as GIS was not really developed until 1963 by Tomlinson.  This may be a problem due to unclear methods.  Were estimates used when known geographic locations were found in the papers (say a collection from a given town in older papers could be assigned current GIS location data for that town in 2021)?  This would allow for older papers to be included more easily. 

-While part of this method was to use the internet search engines to find papers of interest and filter them as in figure 1, this method runs a good chance of passing over older papers which have not been catalogued yet (or improperly).  Part of doing a good review is finding these hidden papers.  Were the search engines used exclusively to find papers, or were other methods included?  I guess the question here is how good a job could the search engines do by themselves, or was this a more traditional review paper (using references within papers to extend the search to previous work and beyond)?

-It is a little shocking to find that over the range of 1913-2021 only 89 papers could be identified as identifying termites collected in Thailand.  There are two ways to take this piece of information.  Were there really 89 surveys of Thailand termite papers (that’s a lot of survey papers)?  Or were other non-survey papers included for example, other studies that identified their subject termites in Thailand?  This more an interesting comment than anything that needs to be corrected in the manuscript.

-While I liked the overall geographical figures, they are difficult to read in their current size.  Older readers may appreciate a larger text, heavier line edging and markers.